# Deep Compressive Sensing on ECG Signals with Modified Inception Block and LSTM

**DOI:** 10.3390/e24081024

**Published:** 2022-07-25

**Authors:** Jing Hua, Jue Rao, Yingqiong Peng, Jizhong Liu, Jianjun Tang

**Affiliations:** 1School of Software, Jiangxi Agricultural University, Nanchang 330045, China; 15870668662@163.com (J.H.); pengyingqiong1@163.com (Y.P.); 2School of Computer and Information Engineering, Jiangxi Agricultural University, Nanchang 330045, China; mussray@163.com; 3Nanchang Key Laboratory of Medical and Technology Research, Nanchang University, Nanchang 330031, China

**Keywords:** deep learning, compressed sensing, ECG signal, LSTM, Inception block

## Abstract

In practical electrocardiogram (ECG) monitoring, there are some challenges in reducing the data burden and energy costs. Therefore, compressed sensing (CS) which can conduct under-sampling and reconstruction at the same time is adopted in the ECG monitoring application. Recently, deep learning used in CS methods improves the reconstruction performance significantly and can removes of some of the constraints in traditional CS. In this paper, we propose a deep compressive-sensing scheme for ECG signals, based on modified-Inception block and long short-term memory (LSTM). The framework is comprised of four modules: preprocessing; compression; initial; and final reconstruction. We adaptively compressed the normalized ECG signals, sequentially using three convolutional layers, and reconstructed the signals with a modified Inception block and LSTM. We conducted our experiments on the MIT-BIH Arrhythmia Database and Non-Invasive Fetal ECG Arrhythmia Database to validate the robustness of our model, adopting Signal-to-Noise Ratio (SNR) and percentage Root-mean-square Difference (PRD) as the evaluation metrics. The PRD of our scheme was the lowest and the SNR was the highest at all of the sensing rates in our experiments on both of the databases, and when the sensing rate was higher than 0.5, the PRD was lower than 2%, showing significant improvement in reconstruction performance compared to the comparative methods. Our method also showed good recovering quality in the noisy data.

## 1. Introduction

Electrocardiogram (ECG) monitoring becomes more and more important in healthcare applications because it can detect the heart disease of the patients as early as possible and ECG signals are often used in many clinical diagnosis, such as identifying arrhythmia, observing the circulation of the blood supply of the coronary artery, etc. However, the ECG data are usually acquired at a high frequency. Furthermore, the data contain long-time records and multiple leads signals which need large storage room to be saved. Thus, there is an urgent need to reduce the burden of data transmission, data storage, and energy consumption. Compressed sensing (CS) is a technology that can conduct signal sampling and reconstruction jointly, and recover signals from fewer measurements than the measurements suggested by the theory of Nyquist–Shannon sampling, namely that the sampling frequency should not be less than two times the highest frequency in the signal spectrum. In this case, CS has the ability to efficiently compress the ECG signals, reducing the amount of data to be transmitted and the energy costs [1]. After the receiving terminal obtains the data, CS reconstructs the signals from the compressed data, which have lower dimensions than the original signals and the reconstructed signals can be applied to the clinical diagnosis in the same way as the original signals [2].

In the traditional CS process, the measurement of the original signal is obtained by multiplying the signal by a known measuring matrix. The reconstruction from the measurement to the original signal is considered as finding a solution in the underdetermined system, when the original signal is sparse in some domains.

The main points of the traditional CS recovery methods are listed below:How to make the signals sparser? In the traditional CS methods, the signals should be originally sparse or become sparse after some transformation. However, not all of the signals are sparse in nature, so the key is to find a proper sparse presentation method. The popular methods mainly include wavelet transform [3], Fourier transform [4], short-time Fourier transform [5], Beamlet transform [6], Curvelet transform [7], Contourlet transform [8], Gabor dictionary [9], K-Singular Value Decomposition (K-SVD) algorithm [10], etc.;How to design a measuring matrix which is easy to realize on the hardware and can satisfy the Restricted Isometry Property (RIP) principle? The RIP principle can be expressed as follows:
(1)$$\left(1-\delta \right){\left\|x\right\|}_{2}^{2}\leq \|\Phi {x\|}_{2}^{2}\leq \left(1+\delta \right){\left\|x\right\|}_{2}^{2}$$
where *δ* is a small constant. For a given measuring matrix Φ and original signal x, if there is a proper *δ* satisfying the Equation (1), we call Φ has RIP property. When the correlation of the matrix is small enough, the measuring matrix Φ has a high probability of satisfying the RIP principle and the signal x can be reconstructed properly. In practice, we usually exploit two methods to generate the measuring matrix used in CS [11]. One is the random measuring matrix, including the Gaussian random matrix, the Bernoulli random matrix, etc. The other is the deterministic measuring matrix, including the Low Density Parity Check (LDPC) code matrix [12], the matrix generated from chaotic sequences [13], etc.;

How to find a good solution to the non-convex optimization problem? Researchers always convert the non-convex optimization problem to the convex optimization problem by changing the objective function or discarding some of the constraints to solve this problem.

Many researchers saw the potential of CS in practical ECG application and they proposed methods to obtain a good performance in the CS process. Zhang et al. [14] proposed the Block Sparse Bayesian Learning (BSBL) algorithm to compress and reconstruct the fetal ECG (FECG) signals with high quality, using the spatial structure and temporal structure of the signals. Izadi et al. [15] exploited the Kronecker technique in the reconstruction phase and applied CS in a very small size to accelerate the compressing phase, resulting in the improvement of the recovered signals. Besides, the adaptive dictionary learning used in reconstruction obtained a better performance than the fixed dictionary. Abhishek et al. [16] developed a novel type of biorthogonal wavelet with a linear phase by exploiting the double-sided exponential splines, which improved the quality of the reconstruction significantly compared to the orthogonal wavelets. Zhang et al. [17] mapped the under-sampled ECG signal onto a two-dimensional space, using Cut and Align to promote the sparsity, and then recovered the signal with a nonlinear optimization model. Melek et al. [18] proposed a novel greedy CS algorithm dubbed ARMP-PKS, using the partial prior support information. They found that even if part of the prior information is not correct, the algorithm can obtain high reconstruction accuracy and speed. Jahanshahi et al. [19] put forward a method with low-rank constraints, using the Kronecker sparsifying base to exploit the temporal and spatial structures of the ECG signals. Saliga et al. [20] adopted a dynamic ECG model in which the parameters were learnt from the measurement by the Differential Evolution algorithm. This model reduced the noise of the interfering signals and maintained the signals’ structures. Rezaii et al. [21] proposed a method dubbed OSOS to determine the number of basis vectors used in the compression process by minimizing the reconstruction error. This method has strong robustness to the noises. The methods mentioned above exploited the sparsity of the ECG signals and usually cannot reach high quality when the sensing rate is rather low. Hua et al. [22] exploited a spatial–temporal sparse model-based method to reconstruct the multichannel ECG signals using the temporal and spatial correlation between the signals from different channels and enhancing the quality with a dictionary-learning algorithm.

Although the traditional CS significantly reduces the data redundancy and reduces the energy costs, there are still some problems in the practical application. The measuring matrix does not contain the structural information of the original signal, thus the reconstruction may not obtain the expected performance, especially when the sensing rate is low. Besides, the storage of the measuring matrix, computation ability and reconstruction time are also the obstacles.

Therefore, with the development of deep learning, the CS recovery of the signals has paved a new way. In the deep network (e.g., Convolutional Neural Network, CNN), all of the process is data-driven and has strong adaptiveness. The original signals need not to be sparse in nature or in k-space, and they can be natural images or medical signals, etc. The measuring matrix is replaced by convolutional layers or full connection layers, and the parameters of the layers can be learnt adaptively. Similarly, the solution to the reconstruction signal is replaced by an end-to-end mapping, which can be learnt automatically during the training of the deep network. In this case, many researchers put forward their methods based on deep learning.

Muduli et al. [23] introduced a method based on the Stacked Denoised Autoencoder (SDAE), to compress and reconstruct the FECG signals and adopted a model similar to the CS-based Multiple Measurement Vector model. The whole network was trained unsupervised with a gradient descent. Polania et al. [24] used Restricted Boltzmann Machines (RBMs) to obtain the probability distribution of the sparsity pattern of ECG signals and reduce the measurement needed in the compression. As a result, the reconstruction accuracy was improved. Mangia et al. [25] estimated the support with an oracle based on a CNN jointly trained with the linear mapping at the decoder. The use of a short-window can significantly reduce the computation complexity. Shrivastwa et al. [26] proposed a CS framework based on CNN to compress and recover the bio-signals (electroencephalogram, EEG). They converted the EEG signals into two-dimensional data, and used the binary image representation of the data to train the model. Mangia et al. [27] exploited an oracle-based CS decoder in an encode–decoder pair to estimate the coefficients by pseudo-inversion. When the compression ratio is higher than 2.5, this method can achieve a good performance in the recovery of the synthetic ECG signals. Shrivastwa et al. [28] used a multi-layer perceptron regressor with a stochastic gradient descent solver to form a CS network which is mainly comprised of full connection layers. Through the iterative boundary values analysis on the size of the network, a network with a better structure can be selected and perform well on the reconstruction.

Compared to the traditional CS methods, the deep learning methods often exploit the relationship between data and data; thus, they have higher reconstruction accuracy. The end-to-end training can remove the dependency of the sparse representation in traditional CS methods, simplifying the process. However, the recovering performance at a low sensing rate is still a problem, so we designed a novel model to improve the quality.

In this paper, we propose a deep-compressive sensing method which is trained end-to end for ECG signals’ compression and reconstruction. Our contributions are listed below:We used three sequential convolutional layers to replace the traditional measuring matrixes to obtain the measurements adaptively, corresponding to the different dimension of original signals. In the convolutional layers, the number of the filters in each layer changed with the sensing rates in the experiment. We compared our compression method with two traditional fixed random matrixes;We exploited a modified Inception block, in which we designed a structure containing a skip connection to use different kernel sizes to extract the features from the signals from different levels. We used the concatenation of those multi-level features to obtain more details of the data, to reconstruct the signals more accurately;ECG signals are time-series signals; thus, we adopted the long short-term memory (LSTM) to deal with the ECG signals. The LSTM is a variant of the Recurrent Neural Network (RNN), and it is appropriate to deal with long sequential signals avoiding the long dependency problems appearing in RNN and the vanishing gradient or exploding gradient in the meantime;We conducted our experiment on two different ECG databases to validate the robustness of our model. We evaluated our methods with other five methods, using the metrics Percentage Root-mean-square Difference (PRD) and Signal-to-Noise Ratio (SNR). From the experiment, we can see that our approach has good quality on both databases. Our methods have higher SNR and lower PRD than other methods, and the reconstructed signals have the best match with the original signals among all of the six methods.

The rest of the paper is organized as follows. Section 2 describes the method we propose, the structure of the network and the details of every module in the network. Section 3 listed details about our experiment and the comparison results of different methods under different sensing rates. Section 4 concludes this paper.

## 2. Background

### 2.1. Compressed Sensing

In compressed sensing theory, the measurements are obtained by multiplication between the measuring matrix and the original signals. The process of compression can be formulated as follows:*y* = Φ*x*(2)
where *y* denotes the measurement; Φ denotes the measuring matrix which has dimension of M × N (M ≪N); and *x* denotes the original signal. Only when Φ satisfies the Restricted Isometry Property (RIP) principle, and *x* is sparse in some domain, can the measurement y be reconstructed accurately, with high probability and quality.

The original signal *x* can be sparsely represented by a set of orthogonal bases $\Psi \;=\left[\psi_{1},\psi_{2},\cdots ,\psi_{N}\right]$. That is,
(3)x=Ψα
where α is the representation of signal *x* under the orthogonal bases Ψ. If α has k (k ≪N) non-zero elements, we regard α k-sparse under the orthogonal bases Ψ. Only when x is sparse or α is sparse, can the measurement be accurately reconstructed.

### 2.2. Deep Learning CS Methods

Many researchers proposed deep learning CS methods, including CNN, Stacked Denoising Autoencoder, Generative Adversarial Network, etc. In this paper, the model is mainly based on CNN, thus we briefly introduce the CS methods based on CNN.

Figure 1 shows a simple example of CNN. In a model, there is often an input layer, an output layer, and many hidden layers corresponding to the different functions and requirements. In the training of the model, the weights and bias of each layer are learnt adaptively, and refined by a back propagation algorithm. In the practical CS application, the original signals after normalization are usually used as an input to the network, and after a series of convolutions, pooling or full connection operations, we can obtain the reconstructed signals as the final output. Different from the classification tasks, the labels of the deep learning CS model are the original signals. Once we choose a proper loss function for the deep network, the final reconstruction can be obtained by minimizing the loss function through back propagation.

## 3. Materials and Methods

In this paper, we propose a deep compressive sensing framework for ECG signals. The model is mainly comprised of four modules: preprocessing; compression; initial reconstruction and final reconstruction. Before they are fed into the model, the original signals should be normalized in the preprocessing module. In the compression module, we can obtain the measurements of the ECG signals through the sequential convolutional layers. In the initial reconstruction module, we raised the dimension of the measurements to the same as the original signals, in order to make the data convenient to deal with in the following process. In the final reconstruction module, we exploited modified Inception block and LSTM to extract the features of the signals in order to further reconstruct the ECG signals with more accuracy. The overall framework is depicted in Figure 2. All of the details above will be described in the following subsections.

### 3.1. Preprocessing

Before the original signal is fed into the model, normalization should be completed. Let $x=\left\{x_{\left(1\right)},x_{\left(2\right)},\cdots ,x_{\left(i\right)}\right\}$ be the original ECG signals, and $x_{\left(i\right)}$ represents the *i*-th signal with 256 sampling points. The signal after normalization ${\hat{x}}_{\left(i\right)}$ can be denoted as follows:(4)$${\hat{x}}_{\left(i\right)}=\frac{x_{\left(i\right)}-min\left(x_{\left(i\right)}\right)}{max\left(x_{\left(i\right)}\right)-min\left(x_{\left(i\right)}\right)}$$ where $min\left(x_{\left(i\right)}\right)$ is the minimum value in $x_{\left(i\right)}$ and $max\left(x_{\left(i\right)}\right)$ is the maximum value in $x_{\left(i\right)}$. After normalization, the values of the input signals locate between zero and one, which contributes to the stable convergence of the model. The training of the model is very important and the process may become slower if the input data are used in the network without normalization.

### 3.2. Compression

Because the fixed measuring matrix used in the CS methods needs a lot of storage space and must satisfy the RIP principle, we used sequential three convolutional layers to replace the fixed measuring matrix in the traditional CS compression process to compress the original signals. The key point is that the convolution can be represented by a multiplication between matrix and matrix [29]. The process can be formulated as:(5)$$y=\sum \left(w_{3}\sum \left(w_{2}\sum \left(w_{1}{\hat{x}}_{\left(i\right)}+b_{1}\right)+b_{2}\right)+b_{3}\right)\;=w_{3}w_{2}w_{1}x+w_{3}w_{2}b_{1}+w_{3}b_{2}+b_{3}$$
where $w_{i}$ and $b_{i}$ (*i* = 1, 2, 3) are the weights and bias, respectively, in *i*-th convolutional layer in the compression module, and *y* is the output (i.e., measurement). The convolution can be expressed as a linear representation of the original signal *x*, and that is why convolution can play the role of measuring matrix in CS.

All of the three convolutional layers have a kernel size of 1 × 4, stride of 4 and filter number of C, where C = 64 × sensing rate (sensing rate equals the ratio of the number of sampling points of measurement and the original signal). Thus, the length of the signals after a convolution operation will drop to a quarter of the length before the convolution. After all of the sequential three convolutional layers, the measurements were obtained, with a size of 1 × 4 × C.

### 3.3. Initial Reconstruction

The size of the measurements is much smaller than the size of the original signals. Before we put the measurements into a final reconstruction process, we had to raise the dimension of the measurements in order to keep the same size as the labels. First, we used a convolutional layer which had a kernel size of 1 × 1, stride of 1 and filter number of 64, followed by a LeakyReLU activation function to map the measurements nonlinearly to the initial reconstruction. After being input into the first convolutional layer in the initial construction, the dimension of the measurements becomes 1 × 4 × 64. Then, the reshape layer can convert the signal with a size of 1 × 4 × 64 into the one with a size of 1 × 256 × 1. The initial reconstruction module ensured that the sizes of input and output were the same, in order to compare the original signal and the reconstructed signal and evaluate the performance of the reconstruction.

### 3.4. Final Reconstruction

#### 3.4.1. Overall Framework

In the final reconstruction module, we first used a 1 × 11 convolutional layer followed by LeakyReLU to convert the signals into data with 16 channels. The reason we chose LakyReLU instead of ReLU is that when the input is negative, the output of ReLU is 0, leading to “Dead Neuron”—a condition under which the parameters cannot be updated. In order to avoid the “Dead Neuron”, the LeakyReLU activation function is a better choice. LeakyReLU can be denoted as follows:(6)$$y_{i}=\left\{\begin{array}{c} x_{i}\;\;\;\;,x_{i}\geq 0\\ \frac{x_{i}}{a_{i}}\;\;\;,x_{i}<0 \end{array}\right.$$
where $x_{i}$ and $y_{i}$ are the input and output of LeakyReLU, and $a_{i}$ is a fixed constant in interval $\left(1,+\infty \right)$. In general, the larger kernel size means larger receptive field and better global features obtained by the convolutions. Through a comparison of the different kernel size, we preferred convolutions with a kernel size larger than 1 × 7, because they can obtain a more accurate construction. Then, we designed a modified Inception block to extract the multi-level features from the signals. Finally, we chose LSTM to deal with the time-series ECG signals to obtain context dependencies among all of the features. In the rest of the section, we mainly introduce the modified Inception block and LSTM.

#### 3.4.2. Modified Inception Block

Inspired by GoogLeNet [30], we adopted the modified Inception block to extract multi-level features from the signals. In the modified Inception block, we used different parallel convolutional layers with kernel sizes of 1 × 7, 1 × 9, 1 × 11, 1 × 13, respectively, in order to obtain information from the different receptive fields.

As shown in Figure 3, the block has four lines, each corresponding to different kernel sizes. In one line, we used skip connection to avoid the network degradation and improve the accuracy of reconstruction [31]. Besides, the skip connection does not increase the parameters in the network or the computation complexity because it only involves the addition of feature maps, rather than multiplication. Thus, the skip connection is appropriate for the signal processing. The skip connection process in one line can be formulated as follows:(7)$$x_{output}^{\left(n\right)}=\sigma \left(f_{2}^{\left(n\right)}\left(\sigma \left(f_{1}^{\left(n\right)}\left(x_{input}^{\left(n\right)}\right)\right)\right)+x_{input}^{\left(n\right)}\right)$$
where $x_{input}^{\left(n\right)}$ and $x_{output}^{\left(n\right)}$ denote the input and output in *n*-th line (*i* = 1, 2, 3, 4), $f_{1}^{\left(n\right)}$ and $f_{2}^{\left(n\right)}$ denote the first and second convolution in *n*-th line (*i* = 1, 2, 3, 4) and σ denotes the LeakyReLU activation function.

Finally, we concatenated all of the outputs of the four lines to obtain a linear combination from them, to form a new feature map with 64 channels in order to fully exploit the information extracted from all of the four lines. This helps promote the cross-channel interaction and the integration of information to reconstruct the signals more accurately. The process can be expressed as follows:(8)$$\;x_{concat}=\left[x_{output}^{\left(1\right)},x_{output}^{\left(2\right)},x_{output}^{\left(3\right)},x_{output}^{\left(4\right)}\right]$$
where the $x_{output}^{\left(i\right)}$ is the output of *i*-th line (*i* = 1, 2, 3, 4), and [·] denotes concatenation operation.

#### 3.4.3. LSTM

LSTM is a recurrent network architecture, with a strong ability to deal with the time-series signals. Unlike in the conventional back propagation through time, the signals flowing in LSTM have a low probability of encountering vanishing gradient or exploding gradient. Besides, LSTM overcomes the long-dependency problem in the traditional RNN network [32].

The structure of the LSTM cell is shown in Figure 4. The LSTM is comprised of an input gate, output gate, forget gate and cell state. The forget gate $f_{t}$ decides what features to use from $c_{t-1}$ to calculate the value of $c_{t}$. The input gate $i_{t}$ is calculated through sigmoid activation function with input data $x_{t}$ and output of hidden node $h_{t-1}$. The update value of the cell state ${\tilde{c}}_{t}$ is determined through a neural network layer, followed by a tanh activation function with $x_{t}$ and $h_{t-1}$. Besides, $i_{t}$ is used to select the features from ${\tilde{c}}_{t}$ to update the value of $c_{t}$. The output gate can be obtained by the similar calculation as the input gate. $h_{t}$ is obtained through the calculation of $o_{t}$ and $c_{t}$. The formulations are listed as follows:(9)$$f_{t}=\sigma \left(W_{f}\cdot \left[h_{t-1},x_{t}\right]+b_{f}\right)$$
(10)$$i_{t}=\sigma \left(W_{i}\cdot \left[h_{t-1},x_{t}\right]+b_{i}\right)$$
(11)$${\tilde{c}}_{t}=tanh\left(W_{c}\cdot \left[h_{t-1},x_{t}\right]+b_{c}\right)$$
(12)$$\;c_{t}=f_{t}*c_{t-1}+i_{t}*{\tilde{c}}_{t}$$
(13)$$o_{t}=\sigma \left(W_{o}\cdot \left[h_{t-1},x_{t}\right]+b_{o}\right)$$
(14)$$\;h_{t}=o_{t}*tanh\left(c_{t}\right)$$
where σ represents the sigmoid activation function, and $W_{*}$ and $b_{*}$ are the weights and bias in the corresponding layers.

## 4. Results

In this section, we first compare our compression method with two fixed random matrices used in traditional CS, proving the performance of our compression module. Then, we compare our CS method with different traditional and deep learning methods, finding that our scheme obtains the best performance in terms of the two evaluation metrics. Finally, we also select another database to validate the performance of the proposed method. The experiment details and comparison results are given below.

### 4.1. Experiment Setting

#### 4.1.1. Dataset

The proposed method is validated using the MIT-BIH arrhythmia database [33]. The database contains 30 min heartbeats records from 48 patients, each with two leads. We choose to use MLII signals for our experiment. Nos. 103, 105, 106, 108, 112, 113, 114, 116, 121, 122, 210, 212, 213, 214, 215, 217, 219, 220, 221 and 222 were chosen as our training datasets, and Nos. 100, 101, 107, 109, 117 as our testing datasets. From the above records, we selected 16,000 ECG signal segments evenly for training and 4000 for testing, each segment with 256 sampling points.

In our experiment, the sensing rate (SR) can be defined as:(15)$$SR=\frac{M}{N}$$
where *N* is the sampling points of the original ECG signal and *M* is the total sampling points of the signal after going through the compression module. In our experiment, *N* is 256 and the sensing rates are 0.05, 0.1, 0.2, 0.3, 0.4, 0.5 and 0.6, respectively.

#### 4.1.2. Evaluation Metrics

In our experiment, in order to measure the difference between the original and reconstructed signal, we used Percentage Root-mean-square Difference (PRD) and Signal-to-Noise Ratio (SNR) to evaluate the performance of all of the approaches. They are formulated as follows:(16)$$PRD\left(\%\right)=\frac{\|x-{\bar{x}\|}_{2}}{{\left\|x\right\|}_{2}}\times 100$$
(17)$$SNR\left(dB\right)=10\log_{10}\frac{{\left\|x\right\|}_{2}^{2}}{\|x-{\bar{x}\|}_{2}^{2}}$$
where *x* and $\bar{x}$ are the original ECG signal and the reconstructed signal, respectively.

PRD describes the error of the reconstruction, and it can be related to the quality of the recovered signals [34]. The smaller the PRD, the better the performance of the reconstruction. The relationship is shown in Table 1. SNR is related to the ratio of the original signal and the noise. The bigger the SNR, the better the reconstruction.

#### 4.1.3. Training Parameters

To calculate the actual difference between the reconstructed signal and the target label, we used the Mean Square Error (MSE) as the loss function to train the network. The loss function can be formulated as follows:(18)$$\;L_{MSE}=\frac{1}{n}\overset{n}{\underset{i=1}{\sum }}\|f\left({\hat{x}}_{\left(i\right)};w,b\right)-{x_{\left(i\right)}\|}_{2}^{2}$$
where f, w and b are the operation, weights and bias of the network, respectively, and n denotes the number of signals in the training. Adam is chosen as our optimizer. We set the initial learning rate to 0.0005, with the first and second moment decay rate as 0.9 and 0.999, respectively. Why we choose 0.0005 as the initial learning rate is because a smaller one may lead to a slower convergence, and a larger one may make the loss function suddenly increase a lot. The training epoch is 200 and the batch size is 32. We conducted our experiment at the sending rates of 0.05, 0.1, 0.2, 0.3, 0.4, 0.5 and 0.6 and used a back propagation algorithm to learn and finetune all of the parameters in the network. Our experiment was implemented on Windows 10, Pytorch 1.9.0, GPU NVIDIA Tesla K80 and CPU Intel Xeon E5-2678 v3.

### 4.2. Comparison of Different Compression Methods

In this paper, we use sequential three convolutional layers to compress the original signals instead of a fixed random matrix. In order to validate our compression module, we compare our compression method with Gaussian random matrix and Bernoulli random matrix, while maintaining the rest of model unchanged. The performance of the methods using our compression method, Gaussian random matrix and Bernoulli matrix is listed in Table 2. From Table 2, we can see that under all of the sensing rates, our compression method outperforms the Gaussian random matrix and Bernoulli random matrix. When the sensing rate is as low as 0.05, the PRDs of the methods using the Gaussian random matrix, Bernoulli random matrix and our compression method are 26.85%, 24.85% and 19.72%, respectively. The SNRs of the methods using the Gaussian random matrix, Bernoulli random matrix and our compression method are 13.08 dB, 13.85 dB and 15.86 dB, respectively. The PRD and SNR of our method is 7.13% lower than and 2.78 dB higher than the method using Gaussian random matrix. Besides, the PRD and SNR of our method is 5.13% lower than, and 2.01 dB higher than, the method using the Bernoulli random matrix. When the sensing rate is 0.4, the PRDs of approaches using Gaussian random matrix, Bernoulli random matrix and our compression method are 3.29%, 3.29% and 1.32%, respectively. The SNRs of the methods using the Gaussian random matrix, Bernoulli random matrix and our compression method are 30.67 dB, 30.90 dB and 38.37 dB, respectively. Our method has the lowest PRD and highest SNR at this sensing rate, and the PRD is lower than 2%, showing “very good” recovering performance. The above results prove that our compression method has a better performance than the Gaussian random matrix and Bernoulli random matrix, because our adaptive measuring scheme can learn the parameters of the filters in the convolutional layers autonomously, therefore the scheme is able to obtain the distribution of data and more details from the original signals, which helps a lot in reconstructing the signals more accurately [35].

The visual reconstruction comparison is depicted in Figure 5. We choose a signal segment about 5 s from No. 109 record in testing datasets, and reconstruct the segment at sensing rate of 0.1. The original signal and the reconstructed signal of different methods are depicted in Figure 5. From Figure 5, we can find visually the reconstructed signal recovered using our method has the best quality, for the reconstructed signal is closest to the original signal among the three methods and the peaks of the reconstructed signal has the best match with the ones of original signal.

### 4.3. Comparison with Other CS Methods

We compare our proposed method with three traditional CS methods (BSBL-BO [36], OMP [37] and SP [38]) and two deep learning CS methods (CSNet [39] and CAE [40]) in terms of PRD and SNR. In OMP and SP, we use random Gaussian matrix as the measuring matrix and in BSBL-BO we use sparse random matrix. In [40], the author reshaped the one-dimension signal into two dimension using zigzag method, thus we convert our signal into shape of 16 × 16 when conducting the CAE experiment.

The average PRDs and SNRs of different CS methods over testing datasets are listed in Table 3. We can observe that at all of the sensing rates, the reconstruction of the proposed method outperforms the other five methods. When the sensing rate is as low as 0.05, the PRDs of BSBL-BO, OMP, SP, CSNet, CAE and our method are 55.69%, 120.10%, 72.82%, 24.89%, 23.38% and 19.72%, respectively. The SNRs of BSBL-BO, OMP, SP, CSNet, CAE and our method are 6.01 dB, −1.09 dB, 4.42 dB, 13.67 dB, 13.99 dB and 15.86 dB, respectively. Our method has the lowest PRD and highest SNR among the six methods. Besides, we can see that the deep learning CS methods have a better performance than the traditional CS methods at a low sensing rate. When the sensing rate is 0.4, the PRDs of BSBL-BO, OMP, SP, CSNet, CAE and our method are 4.40%, 24.54%, 16.51%, 3.18%, 7.87% and 1.32%, respectively. The SNRs of BSBL-BO, OMP, SP, CSNet, CAE and our method are 28.87 dB, 16.29 dB, 19.39 dB, 30.82 dB, 23.31 dB and 38.37 dB, respectively. Our method is the only one which has PRD lower than 2% at this sensing rate, showing the best performance.

Figure 6 shows the trend of PRDs and SNRs using different CS methods at different sensing rates in the MIT-BIH Arrhythmia Database. It can be found in Figure 6 that the PRD curve of our method is the flattest. The PRD of our method is the lowest and the SNR of our method is the highest among all of the six methods at all of the sensing rates in our experiment.

In the same way as in Section 4.2, we chose a signal segment about 5 s from No. 100 record in testing datasets, and reconstructed the segment at sensing rate of 0.1. The original signal and the reconstructed signal of different methods are depicted in Figure 7. We can see from Figure 7 that the reconstruction performance of the traditional CS methods is worse than the performance of the deep learning methods generally. In the three deep learning methods, the peaks of the reconstructed signal are closer to the original signal, and our method has the best performance with the lowest reconstruction PRD of 9.22%.

### 4.4. Experiment on Another Database

To validate the robustness of the proposed model, we choose another dataset—Non-Invasive Fetal ECG Arrhythmia Database [41]—to conduct the experiment once again. This database contains the FECG signals of 26 objects with one lead for the chest signals and four to five leads for the abdomen signals. The ECG channel was used in our experiment. We chose ARR_01-ARR_10 and NR_01-NR_10 records for the training datasets, and ARR_11, ARR_12, NR_11, NR_12 and NR_13 for the testing datasets (ARR records refer to arrhythmia fetus and NR records refer to normal rhythm fetus). In the same way as the experiment conducted on the MIT-BIH arrhythmia database, we select 16,000 ECG signal segments evenly for training and 4000 for testing, each segment with 256 sampling points. The experiment setting differed a little from the one conducted on the MIT-BIH Arrhythmia Database, that is, the initial learning rate was 0.0001 when sensing rate was 0.4, 0.5 or 0.6, in order to foster the model convergence with appropriate step.

The average PRDs and SNRs over the testing datasets in the Non-Invasive Fetal ECG Arrhythmia Database, using different CS methods under different sensing rates, are listed in Table 4. When sensing rate was 0.05, the PRDs of BSBL-BO, OMP, SP, CSNet, CAE and our method are 69.74%, 112.40%, 76.46%, 16.68%, 17.76% and 10.99%, respectively. The SNRs of BSBL-BO, OMP, SP, CSNet, CAE and our method are 3.86 dB, −0.11 dB, 4.13 dB, 17.42 dB, 16.59 dB and 21.32 dB, respectively. The PRD of our approach was the lowest, and the SNR of our method was the highest at this sensing rate. When the sensing rates are 0.5 and 0.6, our proposed method obtained a PRD lower than 2%, implying “very good” recovering performance, while the PRDs of the other comparative methods were higher than 2%, and the PRD of our approach was around 40 dB, far exceeding the other five comparative methods. All of the experiment results mentioned above prove that our scheme has the best reconstruction performance among all of the six methods on the Non-Invasive Fetal ECG Arrhythmia Database.

Figure 8 shows the trend of the PRDs and SNRs using the different CS methods at different sensing rates in the Non-Invasive Fetal ECG Arrhythmia Database. From Figure 8, we can also see that the PRD of our method is the lowest and the SNR is the highest at all of the sensing rates.

We chose a signal segment of about 2 s (the sampling frequency is 1000 Hz) from the ARR_11 record in the testing datasets, and reconstructed the segment at sensing rate of 0.1. The original signal and the reconstructed signal of different methods are depicted in Figure 9. From Figure 9, we can observe that reconstructed signal using our method is closest to the original one, showing the best performance with PRD of 6.52%.

Through the experiments on the two different databases, it can be concluded that our method shows the best reconstruction quality on both of the two databases, proving the robustness of our model.

### 4.5. Performance on Noisy ECG Signals

The previous two databases were noise-free, but, in practice, with electrocardiogram monitoring, there are always noises in the acquired signals, such as motion artifact. To simulate the signals with artifact, we added Gaussian noise into the signals in the training dataset and test dataset in the MIT-BIH Arrhythmia Database with SNR = 32 dB and SNR = 24 dB, respectively. The performance of our method on the noisy data is listed in Table 5. When the dataset is noisy signal with SNR = 32 dB, the PRDs of our method are 19.76%, 11.11%, 5.19%, 3.18%, 2.45%, 2.30% and 2.17% at a sensing rate 0.05, 0.1, 0.2, 0.3, 0.4, 0.5 and 0.6. The SNRs are 15.49 dB, 20.22 dB, 26.29 dB, 30.28 dB, 32.35 dB, 32.85 dB and 33.36 dB. The performance of our method is slightly reduced compared to the performance using a noise-free dataset, with the PRD increasing less than 1.3%. Besides, the PRD is below 9% at a sensing rate over 0.2 and is approaching 2% at a sensing rate over 0.4, showing a good recovery quality. When the dataset is noisy signal with SNR = 24 dB, the signal is mixed with more noise, and the performance obtains a further reduction. The PRDs of our method are 22.71%, 12.73%, 7.78%, 5.99%, 5.25%, 5.11% and 4.79% at a sensing rate 0.05, 0.1, 0.2, 0.3, 0.4, 0.5 and 0.6. The SNRs are 13.81 dB, 18.47 dB, 22.36 dB, 24.52 dB, 25.63 dB, 25.86 dB and 26.42 dB. Although the performance of our method is not as good as our methods using a noisy signal with SNR = 32 dB, the PRD is below 9% when the sensing rate is over 0.2, indicating a good reconstruction.

We chose the same 5s signal segment from record No. 100 in Section 4.3, adding the noise to provide a visual reconstruction presentation. Figure 10 shows the noisy data and the reconstructed noisy data when adding different amounts of noise.

Comparing Figure 10 and Figure 7f, we can observe that, although the performance of our methods obtains degradation on the noisy signal reconstruction in terms of the evaluation metrics, the reconstructed noisy signal maintains the primary structure of the original noisy structure, showing the good learning ability of our method.

## 5. Discussion

From the experiment results, we can find that the deep-learning CS methods gain better performance than the traditional CS methods and the recovered signal of the traditional CS methods is always oscillating. Those traditional CS methods often rely on an iterative optimization algorithm, and their single optimization process does not involve all of the training dataset. Therefore, they cannot capture all of the data distribution at one iteration. In contrast, the deep learning CS methods learn the parameters by the back propagation, using all of the training dataset as the input in one epoch. In this case, the deep learning CS methods have a strong ability to learn the inherent features of the signal and reconstruct the signal with a good quality.

Our method is composed of LSTM, which is able to deal with the ECG signal (time sequential signal) efficiently and has a better performance in extracting the features from the ECG signal than ordinary convolution. The modified Inception block in our scheme extracts features from signal using four parallel lines to integrate the information the four lines obtain and foster the communication between different channels, exceeding the performance of single line. Besides, the compression process use the sequential convolutional layers to compress the signal. The above mention is the key of our better performance than comparative methods. Our future research will focus on how to promote the performance when the sensing rate is rather low.

## 6. Conclusions

In order to improve the recovering performance of the ECG signal in CS manners, we propose a deep compressive sensing framework exploiting the modified Inception block and LSTM. The model mainly includes four modules: preprocessing; compression; and two-stage reconstruction. In this model, we used sequential convolutional layers instead of fixed random matrixes to compress the normalized ECG signals, and used modified Inception block and LSTM in the reconstruction process to recover the signals from the compression. The extensive experiments on the MIT-BIH Arrhythmia Database and Non-Invasive Fetal ECG Arrhythmia Database in our research proves that our compression approach has a better ability to compress the original signal and our scheme outperforms the other five methods in terms of SNR and PRD. The PRD of our proposed method is the highest and the SNR is the lowest of all of the sensing rates in our experiments. When the sensing rate is over 0.5, the PRD is lower than 2% and the SNR is over 39 dB, with the performance significantly exceeding the other methods. Furthermore, when conducting the experiment on noisy ECG signals, our method still obtains a low PRD, showing a good performance.

## Figures and Tables

**Figure 1 entropy-24-01024-f001:**
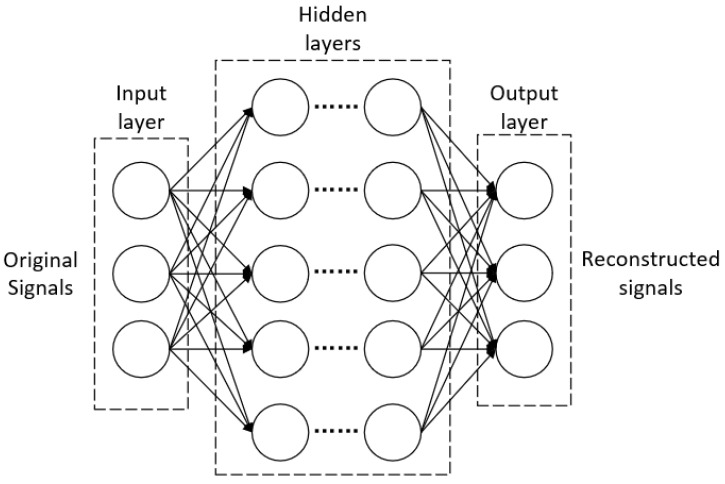
A simple example of CNN.

**Figure 2 entropy-24-01024-f002:**
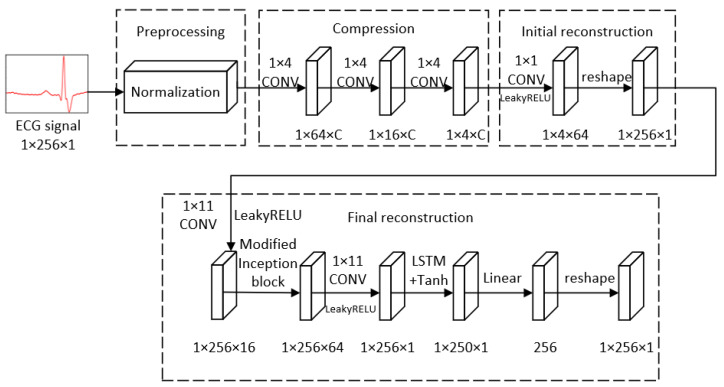
The overall framework.

**Figure 3 entropy-24-01024-f003:**
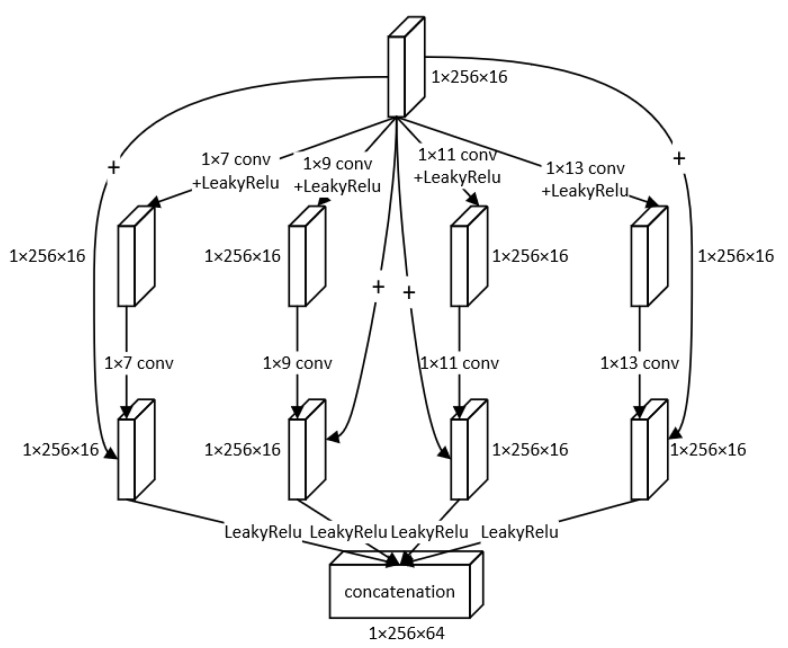
The structure of modified Inception block.

**Figure 4 entropy-24-01024-f004:**
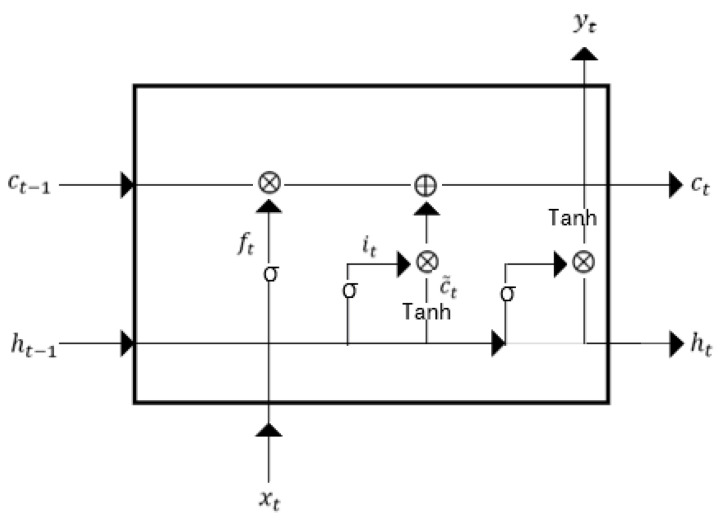
The structure of LSTM cell.

**Figure 5 entropy-24-01024-f005:**
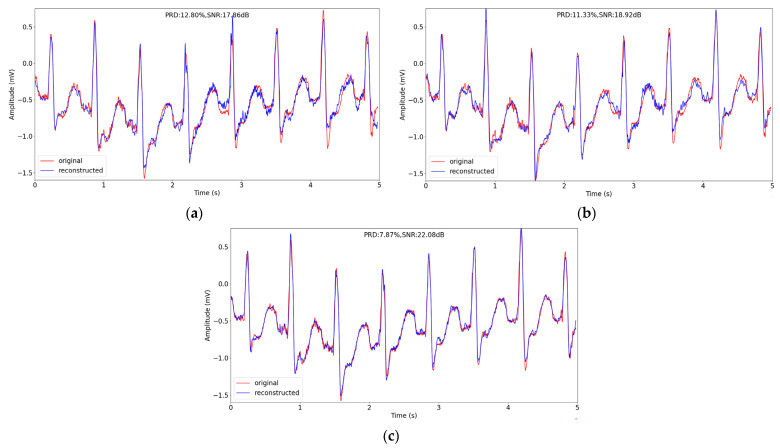
The comparison of original and reconstructed signal about 5 s in record 109 in testing datasets in MIT-BIH Arrhythmia Database using different compression methods when sensing rate is 0.1. (**a**–**c**) correspond to Gaussian random matrix, Bernoulli random matrix and our sequential convolutional layers.

**Figure 6 entropy-24-01024-f006:**
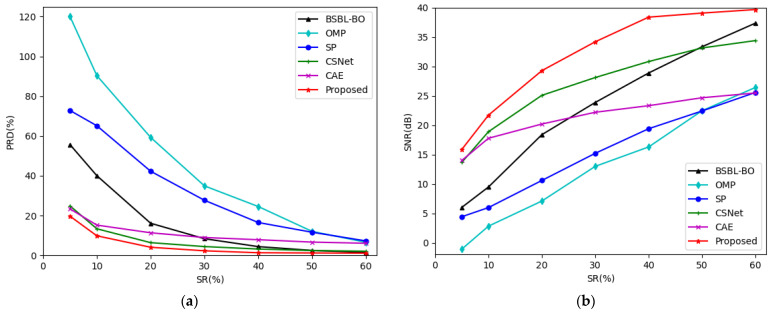
(**a**) The trend of PRDs using different CS methods at different sensing rates in MIT-BIH Arrhythmia Database; (**b**) The trend of SNRs using different CS methods at different sensing rates in MIT-BIH Arrhythmia Database.

**Figure 7 entropy-24-01024-f007:**
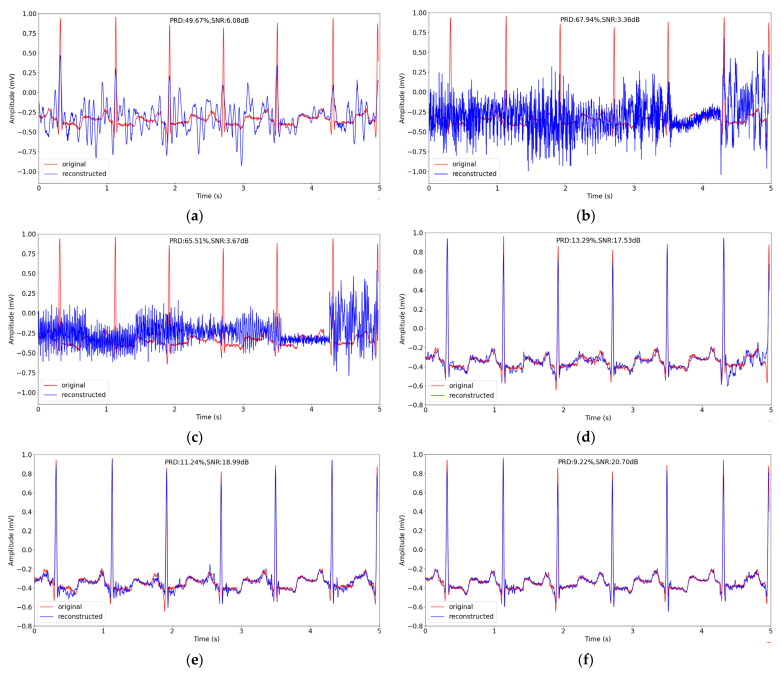
The comparison of original and reconstructed signal about 5 s in record No. 100 in testing datasets in MIT-BIH Arrhythmia Database using different CS methods when sensing rate is 0.1. (**a**–**f**) correspond to BSBL-BO, OMP, SP, CSNet, CAE and the proposed method.

**Figure 8 entropy-24-01024-f008:**
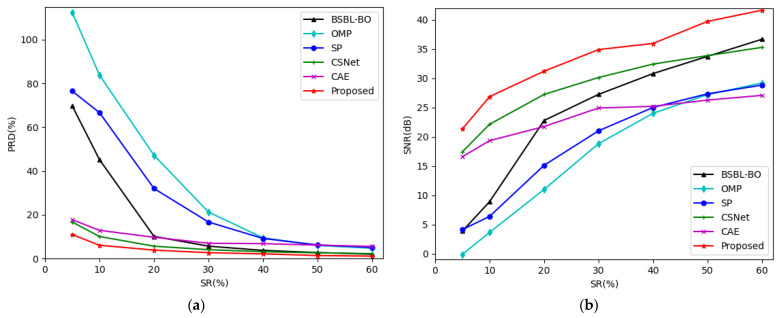
(**a**) The trend of PRD using different CS methods at different sensing rates in Non-Invasive Fetal ECG Arrhythmia Database; (**b**) The trend of SNR using different CS methods at different sensing rates in Non-Invasive Fetal ECG Arrhythmia Database.

**Figure 9 entropy-24-01024-f009:**
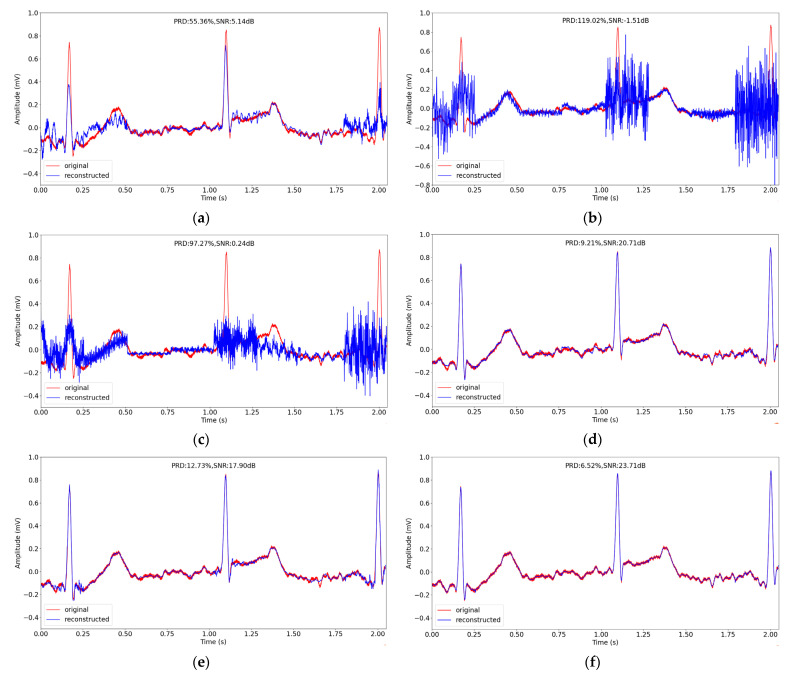
The comparison of original and reconstructed signal about 2s in record ARR_11 in testing datasets in Non-Invasive Fetal ECG Arrhythmia Database using different CS methods when sensing rate is 0.1. (**a**–**f**) correspond to BSBL-BO, OMP, SP, CSNet, CAE and the proposed method.

**Figure 10 entropy-24-01024-f010:**
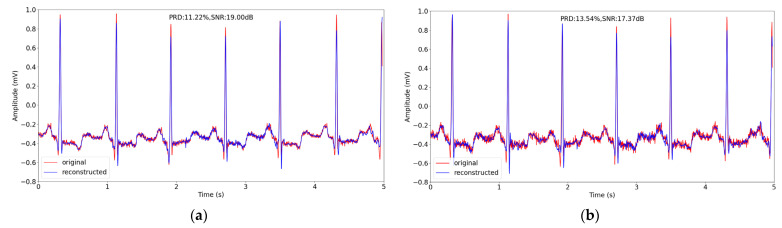
The comparison of noisy signal and reconstructed noisy signal about 5 s in record 100 in testing datasets in MIT-BIH Arrhythmia Database using proposed method when sensing rate is 0.1. (**a**) and (**b**) correspond to noisy signal with SNR = 32 dB and SNR = 24 dB, respectively.

**Table 1 entropy-24-01024-t001:** PRD and the corresponding reconstruction quality description.

PRD(%)	Signals Reconstruction Quality
0–2	Very good
2–9	Good
9–19	No good
19–60	Bad

**Table 2 entropy-24-01024-t002:** Average PRDs and SNRs of different compression methods over testing dataset in MIT-BIH Arrhythmia Database at different sensing rates.

SR	Gaussian		Bernoulli		Proposed	
	PRD	SNR	PRD	SNR	PRD	SNR
0.05	26.85%	13.08 dB	24.85%	13.85 dB	19.72%	15.86 dB
0.1	12.27%	19.62 dB	12.19%	19.61 dB	9.83%	21.70 dB
0.2	6.18%	25.35 dB	6.55%	24.86 dB	4.07%	29.27 dB
0.3	4.25%	28.53 dB	4.66%	27.73 dB	2.29%	34.18 dB
0.4	3.29%	30.67 dB	3.29%	30.90 dB	1.32%	38.37 dB
0.5	2.71%	32.34 dB	2.72%	32.30 dB	1.21%	39.66 dB
0.6	2.44%	33.23 dB	2.44%	33.97 dB	1.12%	39.06 dB

**Table 3 entropy-24-01024-t003:** Average PRDs and SNRs over testing datasets in MIT-BIH Arrhythmia Database at different sensing rates.

SR	BSBL-BO		OMP		SP		CSNet		CAE		Proposed	
	PRD	SNR	PRD	SNR	PRD	SNR	PRD	SNR	PRD	SNR	PRD	SNR
0.05	55.69%	6.01 dB	120.10%	−1.09 dB	72.82%	4.42 dB	24.89%	13.67 dB	23.38%	13.99 dB	19.72%	15.86 dB
0.1	40.04%	9.48 dB	90.33%	2.82 dB	65.14%	5.98 dB	13.36%	18.89 dB	15.21%	17.77 dB	9.83%	21.70 dB
0.2	16.13%	18.38 dB	59.27%	7.09 dB	42.27%	10.59 dB	6.39%	25.07 dB	11.35%	20.19 dB	4.07%	29.27 dB
0.3	8.37%	23.86 dB	34.91%	12.99 dB	27.68%	15.21 dB	4.42%	28.10 dB	8.97%	22.19 dB	2.29%	34.18 dB
0.4	4.40%	28.87 dB	24.54%	16.29 dB	16.51%	19.39 dB	3.18%	30.82 dB	7.87%	23.31 dB	1.32%	38.37 dB
0.5	2.48%	33.33 dB	12.05%	22.45 dB	11.62%	22.41 dB	2.42%	33.13 dB	6.66%	24.66 dB	1.21%	39.06 dB
0.6	1.49%	37.35 dB	6.57%	26.41 dB	7.26%	25.56 dB	2.08%	34.38 dB	6.08%	25.47 dB	1.12%	39.66 dB

**Table 4 entropy-24-01024-t004:** Average PRD and SNR over testing dataset in Non-Invasive Fetal ECG Arrhythmia Database at different sensing rates.

SR	BSBL-BO		OMP		SP		CSNet		CAE		Proposed	
	PRD	SNR	PRD	SNR	PRD	SNR	PRD	SNR	PRD	SNR	PRD	SNR
0.05	69.74%	3.86 dB	112.40%	-0.11 dB	76.46%	4.13 dB	16.68%	17.42 dB	17.76%	16.59 dB	10.99%	21.32 dB
0.1	45.19%	8.90 dB	83.80%	3.64 dB	66.66%	6.38 dB	10.10%	22.14 dB	12.90%	19.33 dB	6.09%	26.87 dB
0.2	10.13%	22.78 dB	47.20%	10.99 dB	31.99%	15.12 dB	5.66%	27.24 dB	9.74%	21.75 dB	3.87%	31.21 dB
0.3	5.68%	27.25 dB	21.22%	18.83 dB	16.70%	21.02 dB	4.07%	30.13 dB	7.00%	24.92 dB	2.70%	34.91 dB
0.4	3.75%	30.81 dB	9.51%	24.01 dB	9.19%	25.01 dB	3.14%	32.43 dB	6.82%	25.20 dB	2.16%	35.96 dB
0.5	2.74%	33.75 dB	5.97%	27.20 dB	6.22%	27.35 dB	2.63%	33.88 dB	6.16%	26.27 dB	1.40%	39.73 dB
0.6	2.01%	36.68 dB	4.74%	29.20 dB	5.17%	28.83 dB	2.27%	35.29 dB	5.56%	27.08 dB	1.12%	41.65 dB

**Table 5 entropy-24-01024-t005:** Average PRD and SNR over noisy testing dataset in MIT-BIH Arrhythmia Database at different sensing rates.

SR	Noisy Data (32 dB)		Noisy Data (24 dB)	
	PRD	SNR	PRD	SNR
0.05	19.76%	15.49 dB	22.71%	13.81 dB
0.1	11.11%	20.22 dB	12.73%	18.47 dB
0.2	5.19%	26.29 dB	7.78%	22.36 dB
0.3	3.18%	30.28 dB	5.99%	24.52 dB
0.4	2.45%	32.35 dB	5.25%	25.63 dB
0.5	2.30%	32.85 dB	5.11%	25.86 dB
0.6	2.17%	33.36 dB	4.79%	26.42 dB

## Data Availability

Publicly available datasets were analyzed in this study. This data can be found here: (https://physionet.org/content/mitdb/1.0.0/, accessed on 24 February 2005 and (https://physionet.org/content/nifeadb/1.0.0/, accessed on 19 February 2019). The source code implementing the proposed method in this paper is available in the url: https://github.com/mussray/DCSECG, accessed on 15 July 2022.

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
