# Peer review of "Deep Compressive Sensing on ECG Signals with Modified Inception Block and LSTM"

_entropy, 2022, doi:10.3390/e24081024_

Round 1
Reviewer 1 Report
The article is interesting and in my opinion valuable. However, it still needs some fine-tuning. I have the following comments:
1. Line 252 - typo “LeakyeLU”.
2. Figures 5, 7 - on the charts axes should be time not samples.
3. Some acronyms need to be explained.
4. The structure of the paper can be confusing, especially in the second half. The authors did not use the section names recommended in the template (Results, Discussion). I would suggest a reorganization. In addition, there is a lack of broader discussion and an attempt to answer the "why" question, not just the "what".
5. The authors selected specific records from the dataset as training and test data. What was the key of these records? The most important here is the test set, for which only 5 records have been selected. In my experience, the results can be significantly different for different records making up the test set. In order to validate the results, I would suggest using cross-validation (e.g. k-fold).
6. The authors used a very complex neural network model. However, they did not write how it was created. Was it the result of testing several models (or other sets of model parameters)? If so, well the results of these tests and alternative models must be presented. Was the model adopted just like that, without deeper analysis? If so, model optimization (e.g., selection of other model structures or metaparameters) must be performed. Is the model an evolution of previously published solutions by the authors. If so, must be cited and compared.
7. It would be valuable to include a link to the source code (GitHub / GitLab) so that the experiment can be repeated.
Reviewer 2 Report
Dear Authors,
In the Line 30 the text remember a bit in context of "shopping" medicine. I would suggest to change it
Reviewer 3 Report
Dear Authors,
Thank you for your contribution. The paper proposes a deep compressive sensing method which is trained on 130 ECG signals used for compression and reconstruction. The article is well written, and the language and structure are good. However, I have some remarks.
In the introduction (line 42) the authors stated that the CS reconstructed signal can be applied for clinical diagnosis. Can you please elaborate more about the point, as it is not clear if the ECG, in general, can be used, or only reconstructed signal? Furthermore, I notice that certain compression and reconstruction techniques compared in the paper are lossy, meaning, that there are parts of information lost during the process. For the clinical diagnosis is crucial that the signal after the processing remains the same or satisfactory for the clinical decision. It is clearly seen that certain methods (not proposed by this paper, but by other authors and compared to the proposed in this paper), perform poorly, and I dought that such reconstructed signal can be used in diagnosis? Can you make discuss such a point? Would you say the proposed method can be used for clinical diagnosis?
>Lines 54-55 "How to design a measuring matrix which is easy to realise on the hardware and can 54 satisfy the RIP principle?"
Please define RIP. I know it is being defined later in the methods, but I suggest that the definition is moved to the introduction session for better readability.
>Lines 100-101 "The original signals need not to 100 be sparse in some domain and they can be anything."
Please, change the phrase "can be anything", as it is unclear. Also, "some" is a weak adjective, so define if it refers to the time/frequency or some other domains?
> Lines 313-314 No.103, 105, 106, 108, 112, 113, 114, 116, 313 121, 122, 210, 212, 213, 214, 215, 217, 219, 220, 221, 222 are chosen as our training datasets, 314 and No.100,101,107,109,117 as our testing datasets
> Lines 428-431 We choose 428 ARR_01-ARR_10 and NR_01-NR_10 records for training datasets, and ARR_11, ARR_12, 429 NR_11, NR_12, NR_13 for testing datasets (ARR records refer to arrhythmia fetus and NR 430 records refer to normal rhythm fetus)
How did you choose the mentioned subjects, was that a random selection, or representative samples? Do you also choose 2sec segment ARR_11, is that subject representative of the other subjects?
I would also like to see how the proposed approach behaves in the presence of some artefacts.
Round 2
Reviewer 1 Report
I am satisfied with the improvements you have made. However, I think that the description of the network that you wrote in response to the review (point 6) should be included in the paper. However, this is my opinion and I leave it to your decision.
In addition, it would be good to add a short description (README.md) with a reference to this article (reference after publication), and translate the Chinese comments in your github repository.
Reviewer 3 Report
Dear Authors,
Thank you for the extensive response and improvement you made to the manuscript.
As I don't have further comments, I can confirm that the authors addressed all the issues, and therefore, I suggest accepting the paper in the present form.